# Higher N Addition and Mowing Interactively Improved Net Primary Productivity by Stimulating Gross Nitrification in a Temperate Steppe of Northern China

**DOI:** 10.3390/plants12071481

**Published:** 2023-03-28

**Authors:** Jianqiang Yang, Huajie Diao, Guoliang Li, Rui Wang, Huili Jia, Changhui Wang

**Affiliations:** 1College of Life Sciences, Shanxi Agricultural University, Taigu 030810, China; 2Shanxi Key Laboratory of Grassland Ecological Protection and Native Grass Germplasm Innovation, College of Grassland Science, Shanxi Agricultural University, Taigu 030801, China; 3Youyu Loess Plateau Grassland Ecosystem Research Station, Shanxi Agricultural University, Taigu 030801, China; 4State Key Laboratory of Vegetation and Environmental Change, Institute of Botany, Chinese Academy of Sciences, Beijing 100093, China

**Keywords:** gross nitrogen mineralization, microbial biomass, mowing, nitrogen deposition, steppe, typical grassland

## Abstract

Anthropogenic disturbance, such as nitrogen (N) fertilization and mowing, is constantly changing the function and structure of grassland ecosystems during past years and will continue to affect the sustainability of arid and semiarid grassland in the future. However, how and whether the different N addition levels and the frequency of N addition, as well as the occurrence of mowing, affect the key processes of N cycling is still unclear. We designed a field experiment with five levels of N addition (0, 2, 10, 20, and 50 g N m^−2^ yr^−1^), two types of N addition frequencies (twice a year added in June/November and monthly addition), and mowing treatment in a typical grassland of northern China. The results showed that higher N addition and mowing interactively improved net primary productivity (NPP), including aboveground and belowground biomass, while different N addition frequency had no significant effects on NPP. Different N addition levels significantly improved gross ammonification (GA) and nitrification (GN) rates, which positively correlated to aboveground net primary productivity (ANPP). However, the effect of N addition frequency was differentiated with N addition levels, the highest N addition level (50 g N m^−2^ yr^−1^) with lower frequency (twice a year) significantly increased GA and GN rates. Mowing significantly increased the GA rate but decreased the GN rate both under the highest N addition level (50 g N m^−2^ yr^−1^) and lower N addition frequency (twice a year), which could improve N turnover by stimulating plant and microbial activity. However, a long-term study of the effects of N enrichment and mowing on N turnover will be needed for understanding the mechanisms by which nutrient cycling occurs in typical grassland ecosystems under global change scenarios.

## 1. Introduction 

Net primary productivity (NPP) in grassland is limited by available soil nitrogen (N) content [1]. The key processes of soil N turnover, including gross ammonification (GA) and gross nitrification (GN), determines soil N availability, which is mainly controlled by the decomposing of soil organic matter from the activities of soil animals and microorganisms [2,3]. Soil gross ammonification and nitrification, defined as the net ammonification and net nitrification gross plus microbial N fixation, which is an important index to characterize the capability of mineral soil supplying for plant growth and microbial reproduction, is thus considered a surrogate of N availability [4]. The conversion of soil N determines plant productivity in terrestrial ecosystems.

Global atmospheric N deposition had increased by 5 to 20 fold since the preindustrial period, with average deposition rates of 2 g m^−2^ yr^−1^ in recent years, and the deposition of reactive N will continue to increase over the next century [5,6,7] Increased N deposition greatly influences the main components of an ecosystem’s N pool (i.e., plant, soil, and microbe N pools) and aboveground net primary productivity (ANPP) [8,9] of grassland ecosystems [9,10,11]. Most studies that simulated N deposition were designed to supply N fertilization once or twice a year [12,13,14], which could not reflect the real, natural N deposition with dry or wet N deposition across the whole year. Moreover, grazing and mowing are the main land-use management mechanisms for maintaining the production and ecological functions of natural grassland ecosystems in northern China. While there were a large quantity of reports of N addition and grazing on soil N turnover by changing nutrient availability and leaching [7,15,16,17], the interactive effects between N deposition and mowing on net primary productivity and the gross N turnover and their relationship grassland ecosystems are still unclear.

The various experimental results from different individual studies were reported in grassland ecosystems that had been conducted to explore how N deposition or fertilization affects N turnover processes in grassland ecosystems, especially in soil N turnover (i.e., N ammonification, nitrification, denitrification, and immobilization) and total N pool [18,19]. Previous studies showed that N addition increased, decreased, or had no effects on soil N turnover [14,20,21,22]. These different results were caused by the N addition time, N addition duration, N types, and the amount of N addition. However, these studies focused on the effects of N input on soil net N mineralization rates, while the responses of soil’s gross N turnover to N enrichment are still limited. The growing season in most of northern natural grassland begins from late April to early May, and the peak growing period of plants is from mid-June to July, thus N addition once or twice a year could not reflect the real effects of the N deposition pattern under natural conditions.

Mowing is one of the most important management mechanisms in the grassland ecosystems of northern China that directly or indirectly affects net primary productivity by influencing microbial activity and N turnover rates [17]. Previous studies reported that mowing also strongly affected nutrient cycling processes because mowing removed aboveground biomass and decreased litter biomass [15,23], which could increase soil surface temperature and decrease soil moisture content and further affect the bacterial and fungi composition and then stimulate N turnover processes [24,25,26,27]. Some studies showed that mowing reduced root biomass and soil total N concentrations [28,29], which was likely due to the loss of N through aboveground biomass harvesting. The change in soil N concentrations would have direct or indirect effects on soil microbial N turnover and aboveground and belowground biomass. However, mowing effects in typical grassland on soil N turnover and the response mechanisms of net primary productivity with the interactions of N deposition remain unclear.

The Inner Mongolia steppe is part of a Eurasian grassland with an area of about 1.18 million km^2^, and it is of great importance for ecological conversation and local farmer livelihoods. We performed an experiment with different N addition levels and frequencies combined with mowing to determine their individual and potential interactive effects on net primary productivity using the soil gross N mineralization rate by the ^15^N pool dilution technique [30] and other N pool components (i.e., plants, soil, and microbes). The gap in knowledge of the interaction between the annual N deposition (i.e., N supply frequency in different months) and mowing effects on the net primary productivity and N turnover processes limits our ability to predict the effects of future climate change on mowing grassland ecosystems. Consequently, our study hypothesis that N input during different months of the whole year could increase plant productivity by improving soil gross N mineralization rates and by stimulating soil microbial activities and could meet the N demand of different plants at different stages.

The objectives of this study are: (i) to investigate how net primary productivity and gross N turnover respond to different levels and frequencies of N addition without or with mowing; and (ii) to definite the relationship between net primary productivity and gross N turnover under different N addition levels and frequencies under mowing treatment.

## 2. Results 

### 2.1. Aboveground and Belowground Biomass

Nitrogen addition treatments, regardless of adding N levels and frequencies, significantly increased aboveground biomass (AGB) and belowground biomass (BGB) (*p* < 0.01). However, mowing showed no significant effects on AGB (*p* > 0.05) but significantly decreased BGB (*p* < 0.001). In addition, the levels and frequency of N addition and mowing significantly and interactively affected AGB and BGB, respectively (*p* < 0.01, Table 1).

Across all N addition levels, AGB increased by 30.92% on average in un-mowing treatments (*p* < 0.05) and increased by 40.80% in mowing treatments (*p* < 0.05) at twice a year of N addition treatment (Figure 1a). Aboveground biomass also increased by 10.67% (*p* > 0.05) and 74.81% in un-mowing and mowing treatments (*p* < 0.05) at higher frequencies (monthly) of N addition, separately (Figure 1a). At the highest N addition level (N-50), the aboveground biomass tended to be higher in mowing versus un-mowing treatments, and AGB at higher N addition levels (N-20, N-50) was significantly higher in mowing than in un-mowing plots at the higher frequency of N addition (*p* < 0.05, Figure 1a). However, no significant effects of the lower frequency of N addition were found on AGB (Figure 1a). Belowground biomass with twice a year of N addition was significantly lower than that at 12 times within a year of N addition under relatively lower N addition levels (N-2, N-10) (*p* < 0.05, Figure 1b), while opposite trends were observed under relatively higher N addition levels (N-20, N-50) (Figure 1b). BGB was significantly increased by 41.46% (*p* < 0.05) under N addition, and it increased with increasing N addition levels at lower N addition frequency (*p* < 0.05).

### 2.2. Gross Ammonification and Nitrification Rates

Nitrogen addition levels and frequencies had significant effects on both gross ammonification (GA, *p* < 0.05) and gross nitrification rates (GN, *p* < 0.05), while mowing showed no significant effects on GA and GN. In addition, the interaction effects of N addition levels and frequencies on GA were significant (*p* < 0.001); however, the effects on GN were not significant (*p* > 0.05). In contrast, the interaction effects between mowing and N addition level or with N addition frequency on GN were significant (*p* < 0.01); however, the effects on GA were not significant (*p* > 0.05, Table 1).

Across all the N addition levels, the highest N addition (N-50) significantly increased GA by 278.45% (*p* < 0.05). For the lowest addition level (N-2), GA was significantly decreased by 44.61% (*p* < 0.05) but was not significantly changed at higher N addition levels (N-10, N-20) with mowing (Figure 2a, Table 1). Higher N addition treatments of N-10 and N-20 significantly increased GN by 398.61% and 548.75% (*p* < 0.05, Figure 2b). Nitrogen addition with mowing significantly increased GN (*p* < 0.05, Figure 2b). However, the interactive effects between frequency of N addition and mowing on GN were not significant (*p* > 0.05, Figure 2b).

### 2.3. The Relationships between Net Primary Productivity and Gross N Turnover

Soil pH values were negatively correlated with soil GA (*R*^2^ = 0.63, *p* < 0.0001) and GN (*R*^2^ = 0.41; *p* < 0.01) (Figure 3a). However, GA (*R*^2^ = 0.87, *p* < 0.0001; *R*^2^ = 0.23, *p* < 0.05) and GN (*R*^2^ = 0.55, *p* < 0.001; *R*^2^ = 0.56, *p* < 0.001) (Figure 3b,c) were positively correlated with the soil NH_4_^+^-N and NO_3_^−^-N concentrations, respectively.

Based on regression analyses, the SEM results indicated that the N addition directly affected N content, soil N, and GA, while it indirectly affected GN and plant biomass through modifying soil N content (Figure 4). Overall, the SEM results suggested that N addition directly affects net primary productivity by increasing inorganic and indirectly affected by increasing gross GA, GN, and the soil N pool, which were the most important factors affecting soil microbial activity.

## 3. Discussion

### 3.1. Aboveground and Belowground Biomass

Our results showed that N input directly increased aboveground biomass in a semi-arid steppe ecosystem, and, on average, N addition significantly increased the aboveground and belowground biomass (AGB and BGB) in this grassland ecosystem; our results were consistent with most studies with N addition experiments [12,31,32,33]. Nitrogen regulates the plant growth and biodiversity of terrestrial ecosystems, and it often determines the net primary productivity of grassland ecosystems [14]. In addition, our results demonstrated that mowing had little effects on AGB and BGB compared with N addition, the level of N addition, or the frequency of N addition. This was due to the fact that N addition significantly changed soil nutrient availability, especially the quantity and quality of available N in the soil, which would have greatly improved the growth of plants. Moreover, mowing significantly decreased vegetation cover, which led to an increase in the surface temperature and a decrease in soil water content. As a consequence, the growth of plants might be restricted in mowing plots, even though N was added [34,35].

The highest NPP and the highest soil GA and GN were found simultaneously in higher N addition levels due to the increased plant coverage and aboveground biomass reduced soil water evapotranspiration, input more organic matter, and improved the soil’s micro-environment by improving soil microbial activity. On the other hand, increased belowground biomass could, effectively, enhance the interception and storage of precipitation [36]. Only positive correlations between GN and ANPP were found. However, we did not find significant correlations between soil GA and ANPP. Our explanation is that GN was the key process in arid and semi-arid grassland that regulated ANPP. On the other hand, ANPP in the natural environment would be affected by different biotic and abiotic factors, such as temperature, soil moisture, and soil fertility [37]. Study found that soil temperature and moisture were the main influencing factors on ANPP during the growing season in Inner Mongolia [38]. Another important reason is that the in situ gross N mineralization rate reflects the instantaneous dynamics under field conditions, and the ANPP is affected by the same external environmental conditions, thus it can establish a closer relationship between gross nitrification and ANPP.

### 3.2. Gross Ammonification and Nitrification Rates

In our study, the degree of the response of gross ammonification rates (GA) to N addition was lower than that of gross nitrification rates (GN) under lower levels of N addition, and GA and GN were strongly affected by the higher levels and frequencies of N addition tested, suggesting that GA and GN were closely related to the concentration of NH_4_^+^-N and NO_3_^−^-N in soils. Few studies have examined how N addition affected gross N turnover in grassland ecosystem. Our results indicated that gross N turnover responded positively to N addition, which was in agreement with numerous studies [39,40]. However, gross N mineralization does not always increase linearly with increasing N deposition levels [33,41,42], in which the rates of gross N mineralization increased up to an intermediate ambient level of N enrichment but then dropped somewhat at N-enriched conditions higher than in these systems. Differences between their results and ours are likely due to differences among the different types of ecosystem studied and environmental conditions. Walecka-Hutchison and Walworth (2007) found that gross nitrification was stimulated by the lower N addition, while it was inhibited by the higher N application, which was inconsistent with our results [43]. Such discrepancies may be caused by the differences of soil nutrient availability, microbial diversities, and soil enzyme activities [44]. In addition, we found that soil pH decreased with increasing levels of N addition, and gross N turnover rates significantly decreased with increasing soil pH. This result was inconsistent with the findings of Cheng et al. (2013) [45], in which GA and GN were positively correlated with soil pH in forests, which was likely due to the differences in the gross N turnover processes foe different ecosystem types, and the major limiting factors were different in forest and grassland ecosystems. Furthermore, many studies indicated that GA and GN were influenced by some other factors aside from soil N availability, e.g., vegetation and soil types [46,47], soil moisture [48], and other human or environmental factors [14,49,50]. However, the studies about the effects of N addition frequency on soil gross N turnover were very scarce, thus further studies are still needed.

### 3.3. Mowing and the Interactive Effects of N Addition and Mowing

Aboveground biomass removal could not only significantly reduce C inputs to soil but also lead to significant N loss, resulting in nutrient limitations to plants and microbes. Consistent with previous reports, we found that mowing had negligible effects on aboveground biomass, GA, and GN, even under different levels of N addition conditions. However, study found that mowing increased nitrification rates and did not affect ammonification and mineralization rates [49]. This difference between our study and theirs may be due to the possibility that the responses of GA and GN to mowing were counteracted by the level and frequency of N addition. On the other hand, mowing might also affect soil gross turnover by changing the environmental conditions (i.e., soil moisture, temperature and so on) in grassland ecosystems [51,52,53]. Another mechanism related to how plant removal might interact with soil N turnover was that the plant material with a large C:N ratio facilitated to rapid N cycling by limiting SOM input to the soil, which would maintain rapid N turnover in the microbial community in an old field in Minnesota, USA. On the other hand, mowing could decrease microbial biomass carbon and N [49], increase soil total C and N by increasing plant diversity [32,54,55], and change the compositions of soil microbial communities [8], which would directly or indirectly impact soil gross N turnover. However, few studies have evaluated the response of gross N turnover to mowing, thus, as a consequence, our knowledge of the response mechanisms involved in such relationships is very limited. Hay harvesting in grassland ecosystems may have significant and direct impacts on exogenous N input to soil microbial N transformation responses to N addition. Aboveground biomass removal by mowing or clipping would mask N fertilizer effects on soil microbial activity by reducing soil organic matter input and root exudation.

## 4. Materials and Methods

### 4.1. Study Site

The study site was located in the Xilin River Basin, a temperate steppe (116°14′ E, 43°13′ N) near the Inner Mongolia Grassland Ecosystem Research Station (IMGERS) in the Inner Mongolia Autonomous Region, China. The experimental area was about 100 ha, which had been fenced since 1999 to exclude large animal grazing. Based on a long-term observational dataset (1980–2013), the mean annual temperature (MAT) was 0.9 °C, with mean monthly temperatures ranging from −21.4 °C in January to 19.7 °C in July, and the mean annual precipitation (MAP) was about 350 mm, with approximately 70–80% happening between May and August. The soil at the site was either a Haplic Calcisol or Calcic-Orthic Aridisol, as classified by the FAO and the US soil classification system, respectively. The dominant plant species were *Stipa grandis* and *Leymus chinensis*, which together accounted for more than 60% of the total peak aboveground biomass, while the others were forbs.

### 4.2. Experimental Design

Eighty experimental plots were established with a randomized complete block design in September 2008 [32]. The area of each plot was 10 m × 10 m, and there were 2 m walkways between any adjacent plots. Five levels of N addition were conducted, namely, 0, 2, 10, 20, and 50 g N m^−2^ yr^−1^ (designated as N-0, N-2, N-10, N-20, and N-50, respectively), with two types of N addition frequencies (twice a year vs. monthly). N additions began on 1 November and the first day of June for the low frequency treatments (F_2_) (2 N addition yr^−1^), or N addition was conducted on the first day of each month for the high frequency treatment (F_12_) (monthly, or 12 N addition yr^−1^), with two management regimes (un-mowing vs. mowing). Hence, there were 20 treatments (5 × 2 × 2) in total, each treatment with 4 replicates. Purified NH_4_NO_3_ (>99%) was used as the N addition treatment. Nitrogen addition was applied referring to the method of Zhang et al. (2016) [32]. Mowing treatments were carried out two times a year, with 10 cm stubble height left after mowing, on 25 June to 30 June and 25 August to 30 August in every year from 2009 to 2018.

### 4.3. Soil and Plant Sampling and Measurements

Plant and soil samples were collected in mid-August 2018. To investigate vegetation biomass and collect plant and soil samples, a 2 m × 0.5 m quadrat was randomly placed in each plot at least 50 cm inside the border of each plot to avoid edge effects. The aboveground biomass (AGB; g m^−2^) was measured by clipping all plants above the soil’s surface, oven-drying the combined clippings at 65 °C for 48 h, recording the sample’s dry weight, and then finely crushing the sample in a mill to determine the C and N content of plants. Three root cores were collected using a 7 cm diameter root auger at 10 cm intervals in the plots to a soil depth of 30 cm. These core samples were then mixed and soaked in water to remove the mineral components of the soil, and then they were treated as described above to measure the belowground biomass (BGB; g m^−2^). Soil samples were collected with a soil augur (3 cm in diameter), and, in each plot, three cores were taken with a 0–10 cm depth at least 50 cm apart. Soil samples were then mixed and sieved through a 2 mm mesh and stored at 4 °C for laboratory analysis of soil ammonium (NH_4_^+^-N; mg kg^−1^), nitrate (NO_3_^−^-N; mg kg^−1^), and moisture (SM%) content, as well as microbial biomass carbon and N (MBC and MBN; mg kg^−1^) content. Soil subsamples were air-dried for analysis of soil pH, soil C, and soil N (C_soil_ and N_soil_; mg kg^−1^). The plant, soil, and root carbon contents were determined by a H_2_SO_4_-K_2_Cr_2_O_7_ oxidation method [56]. The plant, soil, and root N contents were determined by using the Kjeldahl acid-digestion method with an Alpkem auto-analyser (Kjeltec TM 8200, Foss, Denmark). Soil NH_4_^+^-N and NO_3_^−^-N were analysed using a flow injection auto analyser (FLAstar 5000 Analyzer; Foss Tecator, Hillerød, Denmark), in which these substances were extracted from 10 g of fresh soil sample with 50 mL of 0.5 M K_2_SO_4_ solution. MBC and MBN were analysed with a fumigation extraction method [57]. Soil pH was determined by measuring pH of a soil–water suspension (soil:water = 1:2.5) with a pH meter (FE20-FiveEasy, Shanghai, China).

### 4.4. Gross Ammonification and Nitrification Rates

The ^15^N pool dilution technique was used to determine the gross ammonification rate (GA; mg kg^−1^ SDW d^−1^) and gross nitrification rate (GN; mg kg^−1^ SDW d^−1^) in intact soil cores, with one from each experimental plot. Soil in the cores were labelled with ^15^N-(NH_4_)_2_SO_4_ or ^15^N-KNO_3_ (30 atom% ^15^N enrichment and 3 mL 100 g^−1^ dry soil equivalent) for quantification of gross N ammonification or nitrification. A custom-made multi-injector, which consisted of 10 simultaneously operated syringes with custom-made side-port cannulas, was used to ensure homogenous labelling. Each injection was equivalent to 0.1 mL per 100 g^−1^ soil, and then the samples were labelled with 3 mL of ^15^N-enriched solution that corresponded to 2 mg N kg^−1^ dry soil; 3 replicate labels were made from each core. When soil was sampled, the fresh soil core was labelled immediately with the apparatus above mentioned (3 times at 0.3 cm, 1.5 cm, and 2.5 cm depth). About one hour later (t1), the first soil core was collected and extracted by a 0.5 M K_2_SO_4_ solution with a soil/solution ratio of 1:2 as described by Dannenmann et al. (2009) [58]. Forty hours after labelling (t2), the second soil core was extracted using the same procedure as at t1.

Soil extracts were frozen for further processing. The diffusion technique described by Dannenmann et al. (2006) [59] was used to collect NH_4_^+^ and NO_3_^−^ on acid filter traps, which were then analysed for ^15^N enrichment with an isotope ratio mass spectrometer (Delta Plus XP; Thermo, New York, NY, USA) at the Institute of Botany, Chinese Academy of Sciences. Subsamples of the soil extracts were analysed to determine their NH_4_^+^ and NO_3_^−^ concentrations also in laboratory of the Institute of Botany, Chinese Academy of Sciences in order to avoid bias. Gross ammonification and nitrification rates were calculated with the equation provided by Kirkham and Bartholomew (1954) [60].

### 4.5. Statistical Analyses

All results were presented in figures as mean ± SE (standard error). Repeated measures analysis of variance (ANOVA) for a randomized complete block design were performed, and Tukey’s HSD post hoc test was used to test for differences in the response variables between different treatments if the ANOVA revealed an overall significant difference. If *p* < 0.05, differences were concluded to be statistically significant. Linear regression was also performed to examine the relationships between net primary productivity and GA and GN rates with soil pH and NH_4_^+^-N + NO_3_^−^-N. All statistical tests were performed with SAS 9.0, except for regression analyses, which were performed in the SIGMAPLOT software (SIGMAPLOT 12.5 for Windows; Systat Software Inc., San Jose, CA, USA), which was also used to produce graphs. Structural equation modelling (SEM) was carried out with the software package IBM SPSS Amos 21.0.

## 5. Conclusions

Soil N turnover rates and availability are crucial for ecosystem net primary productivity and carbon sequestration, especially in the context of global climate change or anthropogenic disturbances. The frequency of different N addition levels and mowing had only occasional effects on those variables, which was likely due to the effects of the frequency of N addition and mowing being offset by the effects of the levels of N addition, or that interactions of these with other human or environmental factors that we did not examine occurred. Nevertheless, gross ammonification and gross nitrification rates were significantly affected by N addition frequency and mowing, especially at the highest N addition level, which will have feedback to net primary productivity and carbon sequestration. Consequently, long-term research on the response of net primary productivity regulated by N cycling to human activities or environmental variables is needed to further improve our understanding of the response mechanism of N cycling to climatic change.

## Figures and Tables

**Figure 1 plants-12-01481-f001:**
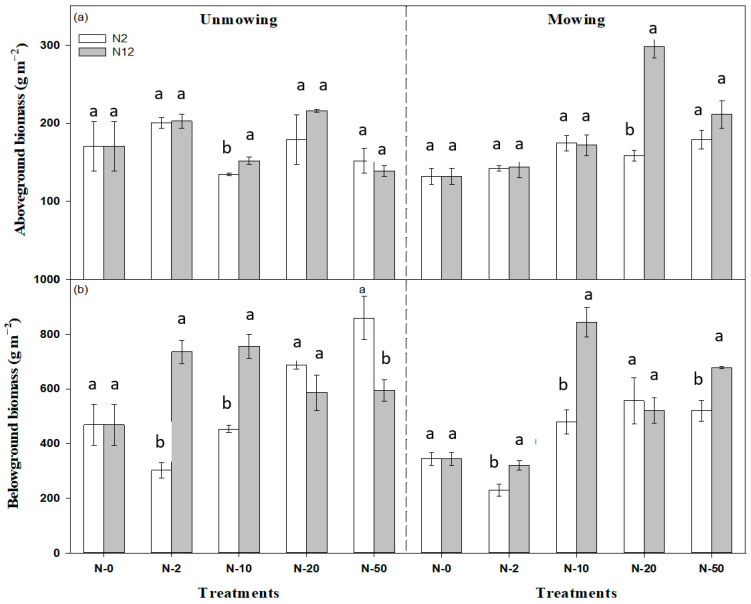
Responses of aboveground biomass (**a**) and belowground biomass (**b**) to N addition rate, N addition frequency, and mowing in the temperate steppe ecosystem studied. The data displayed are the means of four replicates ± standard errors (SE) of the mean. N2 and N12 represent 2 N additions yr^−1^ and 12 N additions yr^−1^, respectively. Different lowercase letters represent groups that are significantly different at the *p* < 0.05 level (Tukey’s HSD test).

**Figure 2 plants-12-01481-f002:**
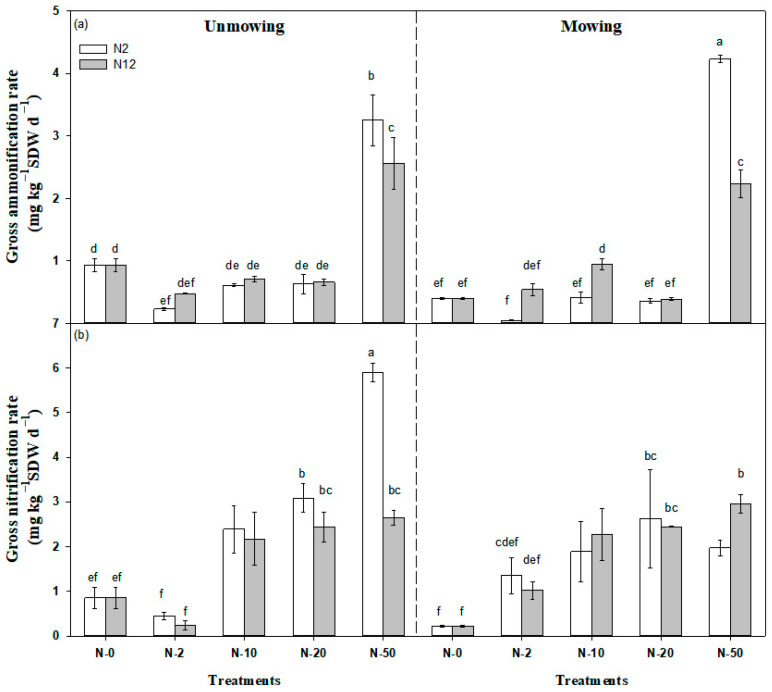
Responses of gross ammonification rate (**a**) and gross nitrification rate (**b**) to N addition level, N addition frequency, and mowing in the temperate steppe. Values are the means of four replicates ± SE (vertical bars). N2 and N12 represent 2 N additions yr^−1^ and 12 N additions yr^−1^, respectively. Different lowercase letters represent groups that are significantly different at the *p* < 0.05 level (Tukey’s HSD test).

**Figure 3 plants-12-01481-f003:**
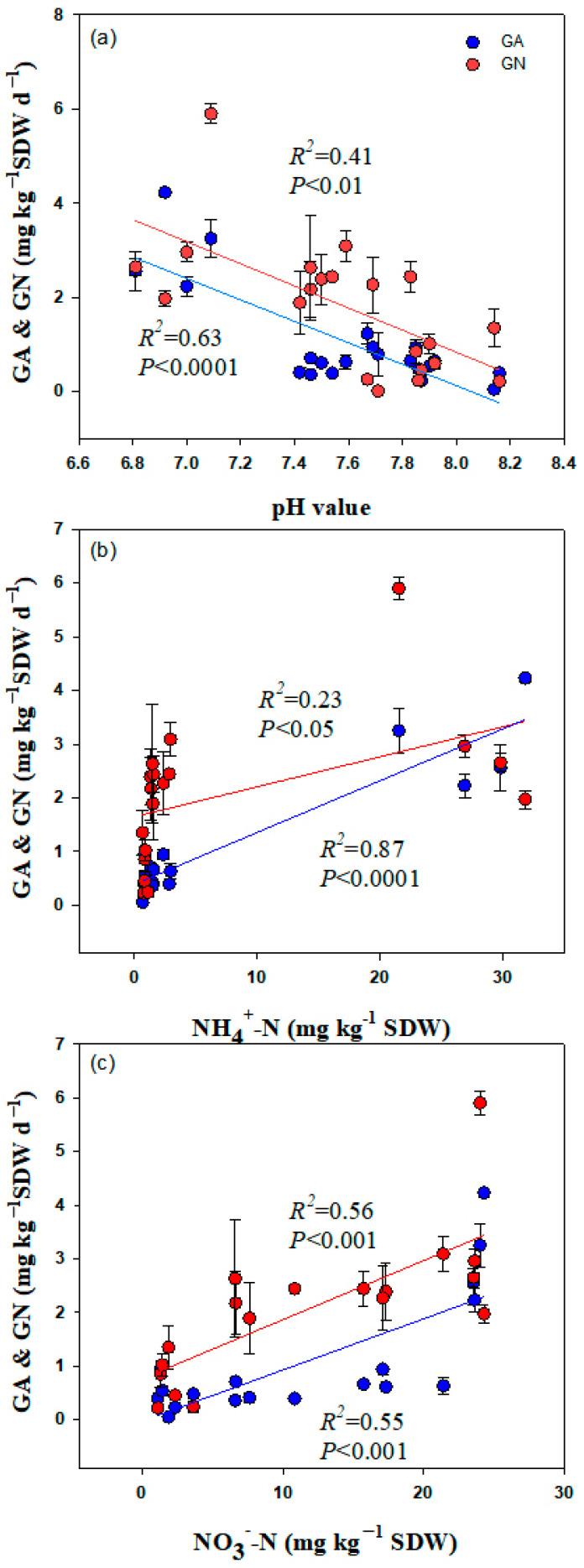
Relationships of gross ammonification rate (GA) and gross nitrification rate (GN) with pH values (**a**), NH_4_^+^-N (**b**), NO_3_^−^-N (**c**).

**Figure 4 plants-12-01481-f004:**
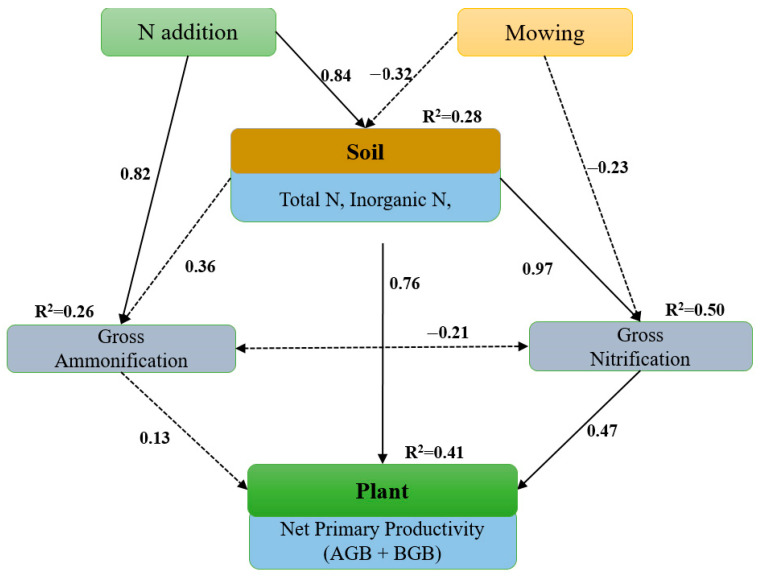
Structural equation model (SEM) analysis of the effects of N addition and mowing on net primary productivity via pathways of soil on gross ammonification rates (GA) and nitrification rates in a temperate steppe in northern China. The bold arrows indicate significant standardized path coefficients (*p* < 0.05).

**Table 1 plants-12-01481-t001:** Results (F values) of repeated measures ANOVAs testing the effects of N addition rate (N), N addition frequency (F), and mowing (M) on soil pH, AGB, BGB, gross ammonification rates (GA), and gross nitrification rates (GN).

	pH	AGB	BGB	GA	GN
N	6.43 ***	11.10 ***	25.16 ***	250.05 ***	25.63 ***
F	0.01	13.74 **	24.51 ***	4.04 *	4.10 *
M	0.22	0.04	20.51 ***	2.22	3.68
N × F	0.21	8.32 ***	19.21 ***	24.25 ***	1.35
N × M	0.24	11.43 ***	6.84 **	2.32	6.22 ***
F × M	0.05	4.88 *	1.15	0.99	8.96 **
N × F × M	0.23	3.68 *	10.61 ***	5.89 **	5.09 **

Note: ‘*’, ‘**’, and ‘***’ represent *p* < 0.05, *p* < 0.01, and *p* < 0.001, respectively.

## Data Availability

The data presented in this study are available on request from the corresponding author (E-mail: wangch@ibcas.ac.cn).

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
