# Peer review of "Higher N Addition and Mowing Interactively Improved Net Primary Productivity by Stimulating Gross Nitrification in a Temperate Steppe of Northern China"

_plants, 2023, doi:10.3390/plants12071481_

Round 1

Reviewer 1 Report

I read the manuscript, and I consider that the topic is falling into the scope of the journal and could be of interest for the readers of the journal. Mowing is one of the most important management measures for temperate grassland in Inner Mongolia. It not only affects grassland productivity, but also affects the carbon and nitrogen cycle of grassland. Unreasonable utilization caused serious grassland degradation. Studying mowing and nitrogen addition is of great significance to grassland restoration. But I think there are some small format problems and details that need to be further improved. For example:

L93: hm2; Some superscript and subscript problems of unit numbers;

L256: NH4+-N, and NO3—N

The bracket problem of 208 lines

In the content of the experimental design to further clarify the fertilization time, two times a year respectively when

What does the significant letter represent in the statistical analysis of Figure 1 and Figure 2, now seems a little unclear. For example, the first graph of Figure1 a does not have the letter a, directly out of the letters b, c, d, so that difficult to understand. Please verify.

Author Response

Dear Editor,

Thank you for handling our manuscript. We are pleased to receive all the positive and constructive comments from three reviewers. Accordingly, we have revised the manuscript to address the comments raised by reviewers. And the “Elsevier Language Editing services” was also used for a further language modification of this manuscripts. Below, we provided a detailed point by point response to their comments.

Sincerely

Changhui Wang

Review 1#

I read the manuscript, and I consider that the topic is falling into the scope of the journal and could be of interest for the readers of the journal. Mowing is one of the most important management measures for temperate grassland in Inner Mongolia. It not only affects grassland productivity, but also affects the carbon and nitrogen cycle of grassland. Unreasonable utilization caused serious grassland degradation. Studying mowing and nitrogen addition is of great significance to grassland restoration. But I think there are some small format problems and details that need to be further improved. For example:

L93: hm2; Some superscript and subscript problems of unit numbers;

Response: Thanks a lot for point out this, we have corrected it and checked the whole text of these problems.

L256: NH4+-N, and NO3—N

Response: Thanks a lot for point out this, we have corrected it

The bracket problem of 208 lines

Response: Thanks a lot for point out this, we have corrected it

In the content of the experimental design to further clarify the fertilization time, two times a year respectively when

Response: Thanks a lot. We have added the information to the material and method section:

 N additions started on 1 November and were continued on the first day of June of the same year and November of the next year for the low frequency treatments (F2) (2 N addition yr-1), or started on 1 November and continued on the first day of each month for the high frequency treatment (F12) (monthly, or 12 N addition yr-1).

What does the significant letter represent in the statistical analysis of Figure 1 and Figure 2, now seems a little unclear. For example, the first graph of Figure1 a does not have the letter a, directly out of the letters b, c, d, so that difficult to understand. Please verify.

Response:

Thanks a lot for point out this. We have added the illustration in to figure legends. “Bars with different lower-case and upper-case letters are significantly different between the F2 and F12 within one N rate treatment are indicated by p < 0.05.

Review 2#

Comments and Suggestions for Authors

Overall, the authors designed and completed a very solid study and I think the questions addressed of impacts of fertilization frequency as well as interactions between N deposition and grassland management are ones of great interest. I was a little confused by the implication that mowing included the removal of biomass. I assumed mowing was simply cutting back the above ground biomass and allowing the cut biomass to fall in place. If this was a case of cutting and collecting the biomass I think harvesting would be a better term.

Response: Thanks a lot, we have removed biomass. We total agree with the better term of “harvesting”

My only suggestion for improving the paper is to have a native English speaker or professional editor review the text, particularly the abstract and the introduction where there were some incomplete sentences and verb tense mismatches.

Response: Thanks a lot. We have invited a native English speaker to revise the paper and hope it has been improved.

Reviewer 2 Report

Overall, the authors designed and completed a very solid study and I think the questions addressed of impacts of fertilization frequency as well as interactions between N deposition and grassland management are ones of great interest. I was a little confused by the implication that mowing included the removal of biomass.  I assumed mowing was simply cutting back the above ground biomass and allowing the cut biomass to fall in place.  If this was a case of cutting and collecting the biomass I think harvesting would be a better term.

My only suggestion for improving the paper is to have a native English speaker or professional editor review the text, particularly the abstract and the introduction where there were some incomplete sentences and verb tense mismatches.  

Author Response

(The authors gave the same response as above.)
